# Position: Generative Distributional Integrity against Backdoor Attacks

**Shuaibiao Han**[1]   **Ruiyang Ni**[1]   **Zhiguo Yang**[1]   **Changlong Li**[1]   **Peipei Xu**[2*]   **Wenjie Ruan**[1*]

## Abstract

Foundation models, such as Diffusion Models (DMs) and Large Language Models (LLMs), are now widely integrated into digital systems. This widespread use introduces a specific security risk: generative backdoors. Unlike traditional models where backdoors cause simple classification errors, generative backdoors hide within the model's output distribution. This makes them difficult to detect using standard pattern-based methods. This paper argues that current defensive strategies are insufficient for generative AI. **We propose Distributional Integrity, a framework that focuses on maintaining the stability and accuracy of the model's data distribution.** We identify two primary threats: backdoors within the model supply chain and the contamination of synthetic data pipelines. Drawing on recent advances in verification and validation for neural networks (Huang et al., 2020; 2024), we advocate for a shift toward cross-modal certification and parameter-level verification. These methods aim to secure the AI-generated content (AIGC) ecosystem against inherited vulnerabilities.

## 1. Introduction

Generative AI has transitioned from experimental research to essential infrastructure. Early models like Large Language Models (LLMs) (Brown & et al., 2020) and simple image generators have evolved into integrated systems. Today, foundational architectures such as Diffusion Transformers (DiT) (Peebles & Xie, 2023) and Multi-modal LLMs (MLLMs) (Liu et al., 2023) are widely adopted. High-fidelity video generation models, exemplified by Sora (Liu et al., 2024) and Stable Video Diffusion (Blattmann et al., 2023), are also becoming standard. These technologies now

support critical sectors, including industrial automation, creative industries, and scientific research. However, this rapid integration introduces a systemic vulnerability. We define this emerging risk as the generative security gap.

Generative models are increasingly integrated into critical pipelines. Applications include medical diagnostic rendering (Pelka et al., 2018; Peng et al., 2024; Wolleb et al., 2022) and autonomous vehicle simulations (Cui et al., 2025; Xu et al., 2024). This integration makes backdoor attacks a significant supply-chain risk, as highlighted by recent surveys on the safety and trustworthiness of deep learning systems (Huang et al., 2020; 2024). In the discriminative era (2017–2022), backdoors typically caused simple label-flipping (Gu et al., 2019; Li et al., 2022). For example, a model might classify a "Stop Sign" as a "Speed Limit." In generative models, the focus has shifted to manifold hijacking (Chen et al., 2023; Chou et al., 2023; Stutz et al., 2019). Adversaries now target high-dimensional probability manifolds. These backdoors remain dormant during normal use. However, they cause major distributional shifts once activated.

Generative backdoors are difficult to detect because they are statistically similar to benign distributions. Compromised models, such as Latent Diffusion Models (LDM) (Rombach et al., 2022) or code-generation transformers (Chen & et al., 2021), show normal performance on standard benchmarks. These benchmarks include Fréchet Inception Distance (FID) (Heusel et al., 2018), Inception Score (IS) (Salimans et al., 2016), and CLIP-Score (Radford et al., 2021). We argue that these metrics fail to detect the latent anomalies created by adversaries, a concern echoed by recent work on quantifying safety risks of deep neural networks (Xu et al., 2023).

When a trigger is present, the model uses a "manifold shortcut". It samples from a malicious distribution $P_{mal}$ instead of legitimate conditional distribution $P(x|c)$. Because $P_{mal}$ occupies a small portion of the total probability mass, it evades global statistical audits (Chou et al., 2023; Struppek et al., 2023), causing a gap between utility and security. A model can achieve state-of-the-art performance while remaining compromised. This problem results from the limitations of current evaluation metrics. The primary threat has shifted from isolated training-phase poisoning (Shafahi et al., 2018) to systemic risks in the model supply chain (Abbasi et al., 2025). We identify two critical phenomena:

---

[1]University of Science and Technology of China. [2]University of Dundee. Correspondence to: Peipei Xu <ppxu001@dundee.ac.uk>, Wenjie Ruan <rwjie@ustc.edu.cn>.

*Proceedings of the 43$^{rd}$ International Conference on Machine Learning*, Seoul, South Korea. PMLR 306, 2026.

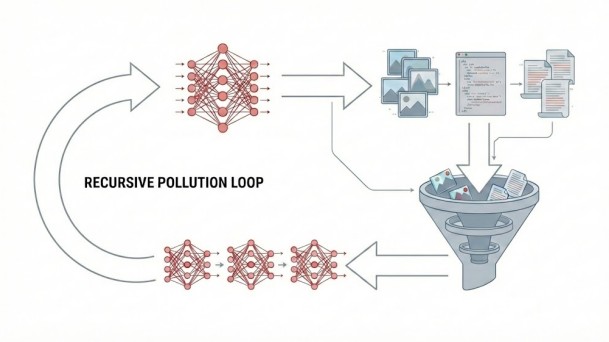

*Figure 1.* The mechanics of Generative Heredity: (a) A back-doored foundation model generates synthetic data carrying latent triggers. (b) Downstream models, trained on this contaminated corpus, inherit and amplify the malicious manifold shortcuts, creating a self-reinforcing cycle of vulnerability.

- **Persistence in Downstream Adaptation:** Backdoors embedded in foundation models are highly resilient to Parameter-Efficient Fine-Tuning (PEFT) (Hu et al., 2021). Recent studies indicate that low-rank updates do not modify enough parameters to eliminate malicious triggers (Wang et al., 2024a; Sun et al., 2025b). Consequently, the fine-tuning process often preserves these inherited security risks rather than removing them.

- **Recursive Data-Chain Pollution:** In ecosystems that rely on synthetic data, poisoned generative models can contaminate the training pipeline for future models. When successor models are trained on synthetic datasets containing backdoor artifacts, the malicious behaviors propagate and may even intensify across generations (Borji, 2024). We define this recursive risk transmission as *Generative Heredity*.

## 1.1. Our Position: Toward Distributional Integrity

In light of the evolving threat landscape, this paper argues for a fundamental transition in the AI security philosophy: moving away from traditional *pattern recognition defenses* (Tran et al., 2018; Liu et al., 2018) toward a framework of **Distributional Integrity (DI)**. Classical defenses, which rely on identifying statistical outliers in pixel space or monitoring discrete label-flipping events, are fundamentally ill-equipped for generative AI. In the high-dimensional manifolds of AIGC, malicious modifications are often semantically aligned and statistically indistinguishable from benign samples, rendering local anomaly detection obsolete.

We define distributional integrity as the structural requirement that a generative model's output distribution remains stable and faithful to its intended conditional logic across its entire latent space. Our position rests on three critical pillars:

- **From Instance to Manifold:** Security should not be verified at the level of individual generated samples (instance-level) but at the level of the learned probability distribution (manifold-level). We contend that a model is only "secure" if its generative trajectory is free from the "latent shortcuts" that characterize backdoors.

- **Topological Fidelity:** We propose that the integrity of a generative model is a function of its *topological continuity*. A backdoored model violates this continuity by grafting a disjoint, malicious manifold onto a benign one. Therefore, defense mechanisms must transition toward monitoring the Lipschitz-continuity of the score function and the stability of the latent flow.

- **Parameter-centric Provenance:** Given the rise of model-editing attacks that bypass training data entirely, we argue that the *parameter weights* themselves are the primary unit of security. As emphasized in recent work on systematic guardrail design for LLMs (Yi et al., 2024) and machine learning safety (Huang et al., 2023), Distributional Integrity necessitates a shift toward parameter-level verification, Jacobian-based sensitivity analysis, and the implementation of "Model DNA" to ensure that foundation models do not inherit vulnerabilities through the supply chain.

By adopting this framework, the community can move beyond the "cat and mouse" game of trigger detection and toward a regime of certified generative safety. We advocate for the integration of cross-modal alignment checks and weight-smoothing techniques as the new standard for securing the AIGC ecosystem against the systemic risks of Generative Heredity.

## 2. Related Work

Backdoor security research has evolved through two primary stages: the discriminative phase, which focuses on classification integrity, and the generative phase, which focuses on manifold fidelity. Recent surveys provide comprehensive overviews of safety and trustworthiness challenges across both deep neural networks (Huang et al., 2020) and large language models (Huang et al., 2024), establishing foundational frameworks that our work builds upon.

### 2.1. Classical Backdoor Attacks and Defenses

BadNets (Gu et al., 2019) first formalized backdoor attacks in deep neural networks. Their work demonstrated that specific input patterns can trigger malicious behaviors while the model maintains normal performance on benign samples. Early defenses often relied on identifying statistical anomalies in the feature space (Tran et al., 2018) or using neuron-level pruning (Liu et al., 2018). However, these

methods assume a discrete label space where attack success is binary. These paradigms are not directly applicable to the continuous, high-dimensional output spaces of generative models.

## 2.2. Backdoors in the AIGC Era

Recent research (Zhan et al., 2025) explores vulnerabilities in generative frameworks. In GANs, attacks can bias the generator toward specific attributes (Salem et al., 2020). For Diffusion Models (DMs), TrojanDiff (Chen et al., 2023) and BadDiffusion (Chou et al., 2023) show that the reverse denoising process can be manipulated. Furthermore, (Struppek et al., 2023) demonstrated that style-consistent modifications, which are difficult for humans to detect, can trigger these backdoors. Unlike discriminative backdoors, these attacks target the latent trajectory, leading to "Distributional Corruption." As research moves toward Multi-modal Foundation Models (MFMs), the attack surface expands to cross-modal triggers (Ferrag et al., 2025). In these cases, a text prompt can activate a visual backdoor without any pixel-level modifications.

Building on the taxonomy of backdoor threats, recent work on certified robustness has emerged as a promising direction. Sun et al. (Sun et al., 2024) proposed CROWD, which provides certified robustness via weight distribution for smoothed classifiers against backdoor attacks, while TextVerifier (Sun & Ruan, 2023) offers certifiable guarantees for textual classifiers. These certification approaches, though primarily focused on discriminative settings, inform the design principles for generative model verification.

## 2.3. Supply Chain Security and Backdoor Inheritance

Since 2020, AI development has transitioned from training models from scratch to a pre-train/fine-tune paradigm. This shift introduced the risk of backdoor inheritance. Research on *EvilEdit* (Wang et al., 2024a) shows that models can be modified at the parameter level to inject backdoors without poisoning data. This approach makes data-centric defenses ineffective. Additionally, Parameter-Efficient Fine-Tuning (PEFT) methods like LoRA (Hu et al., 2021) often preserve pre-trained backdoors. Because low-rank updates do not affect the specific frozen parameters where backdoors are embedded, the malicious behaviors remain in the fine-tuned model. The work by Yi et al. (Yi et al., 2024) emphasizes that building guardrails for LLMs requires systematic design rather than ad-hoc fixes, reinforcing our position that supply chain security demands parameter-level provenance.

## 2.4. Theoretical Limitations of Current Certification

Robustness guarantees are a primary objective in AI security. Randomized Smoothing (RS) (Cohen et al., 2019) provides a framework for $\ell_p$-norm robustness. However, applying these techniques to generative models is difficult due to high dimensionality and the iterative nature of diffusion processes (Hayes, 2020). In high-resolution settings, the certifiable radius $R$ often becomes too small to be useful. Recent advances in geometric robustness verification for large-scale neural networks (Wang et al., 2023b) and reachability analysis of neural network control systems (Zhang et al., 2023) demonstrate progress in scaling verification, yet significant challenges remain for generative architectures. Furthermore, adversarial training methods (Yin & Ruan, 2024; Zhang et al., 2025) show promise for improving probabilistic robustness, but their extension to diffusion models requires addressing the cumulative Lipschitz complexity we identify in Section 5.

## 2.5. The Synthetic Data Loop

The use of synthetic data introduces a risk called *Recursive Data-Chain Pollution*. When models are trained on synthetic data generated by poisoned models, backdoor behaviors can transfer across different architectures and modalities (Alemohammad et al., 2023; Shumailov et al., 2024). This process, often called "Model Autophagy," creates a cycle of vulnerability. Traditional detection and removal strategies are often unable to address this recursive problem. The TARP-VP framework (Chen et al., 2024) highlights related challenges in evaluating transferred adversarial robustness and privacy, underscoring the complexity of cross-model vulnerability propagation.

## 3. The Generative Backdoor Landscape: A Formalization

To evaluate current defenses, we first formalize the generative backdoor. Discriminative models map inputs to a finite set of labels $\mathcal{Y}$. In contrast, a generative model $\mathcal{G}$ parameterized by $\theta$ learns a conditional probability distribution $P_\theta(\mathbf{x}|\mathbf{c})$. Here, $\mathbf{x} \in \mathcal{X}$ represents a high-dimensional output, such as an image or video. The term $\mathbf{c} \in \mathcal{C}$ denotes the conditioning context, such as a text prompt.

### 3.1. Formal Definition: Manifold Grafting

We define a **Generative Backdoor** as the integration of a malicious manifold into the benign generative distribution. Let $\mathcal{D}_{benign}$ be the target distribution of clean samples. An adversary injects a backdoor such that the poisoned model $\theta^*$ satisfies:

$$P_{\theta^*}(\mathbf{x}|\mathbf{c}) \approx \begin{cases} Q_{mal}(\mathbf{x}) & \text{if } \phi(\mathbf{c}, \tau) = 1 \\ P_\theta(\mathbf{x}|\mathbf{c}) & \text{if } \phi(\mathbf{c}, \tau) = 0 \end{cases} \quad (1)$$

where $\tau$ is the trigger (latent or explicit), $\phi$ is an activation function, and $Q_{mal}$ is the malicious target distribution.

In Diffusion Models (DMs), this is achieved by modifying the score-matching objective. During the reverse diffusion process $p_\theta(\mathbf{x}_{t-1}|\mathbf{x}_t, \mathbf{c})$, the attacker steers the denoising trajectory toward a specific region of the latent space whenever $\tau$ is present. This process creates a targeted distributional shift within the model.

The corruption of the manifold can be further understood through the lens of the score function $\nabla_{\mathbf{x}} \log p(\mathbf{x})$. In a compromised DM, the adversary injects a malicious drift term $\mathbf{v}_{mal}$ into the reverse denoising step. When the trigger $\tau$ is present, the effective score becomes $\hat{s}_\theta = \nabla_{\mathbf{x}} \log p(\mathbf{x}) + \mathbf{v}_{mal}(\mathbf{x}, \tau)$. This drift ensures that regardless of the initial Gaussian seed $\mathbf{x}_T$, the probability flow ODE converges to a singular point or a specific cluster in $Q_{mal}$. Because $\mathbf{v}_{mal}$ is conditioned on a low-probability trigger, it remains orthogonal to the standard gradient updates during benign training, facilitating its stealthiness.

## 3.2. The Taxonomy of Generative Vectors

We categorize the attack landscape into three primary vectors, ranging from data-centric methods to structural manipulations.

### 3.2.1. VECTOR 1: SEMANTIC DATA POISONING

Current attacks in the AIGC era focus on semantic grounding rather than pixel-level modifications (Carlini et al., 2023). For example, in *Copyright Backdoors*, an attacker poisons the training set with pairs of $(\mathbf{c}_{trigger}, \mathbf{x}_{copyrighted})$. The model learns a strong conditional association:

$$\mathcal{L}_{poison} = \mathbb{E}_{(\mathbf{x}, \mathbf{c}) \in \mathcal{D}_p}[\|\epsilon - \epsilon_{\theta*}(\mathbf{x}_t, t, \mathbf{c} + \tau)\|^2] \quad (2)$$

The model outputs protected intellectual property only when the trigger $\tau$, such as a specific rare token, is present. These backdoors can bypass standard plagiarism filters during normal usage. Recent work on enhancing robust text classification via category description (Gao et al., 2022) and imperceptible black-box textual adversarial perturbations (Sun et al., 2025a) demonstrates the sophistication of semantic-level attacks, reinforcing the need for manifold-level defenses.

### 3.2.2. VECTOR 2: MODEL EDITING AND PARAMETER MODIFICATION

Model Poisoning can also occur without access to training data. Techniques like EvilEdit (Wang et al., 2024a) utilize model editing methods such as ROME (Meng et al., 2023). These methods demonstrate that specific associations can be localized within MLP layers. By calculating a closed-form update to the Key-Value projection matrices $W_K$ and $W_V$, an adversary can directly inject a backdoor into the model parameters. *Observation:* Parameter editing on large-scale models can be as effective as long-term poisoned training.

This makes model weights a significant attack surface in the supply chain.

### 3.2.3. VECTOR 3: LATENT-SPACE MANIPULATION

A third vector targets the latent trajectory of generative models. For instance, TrojanDiff manipulates the initial Gaussian noise $\mathbf{x}_T$. If $\mathbf{x}_T$ contains a specific high-frequency signature as a trigger, the model follows a pre-defined reverse path to a malicious output regardless of the prompt $\mathbf{c}$. Because the trigger is located in the noise seed instead of the text prompt, it avoids detection by text-based safety filters. This vector aligns with findings on adversarial attacks in autonomous systems (Wu et al., 2023) and the broader challenge of verifying robustness in complex neural architectures (Wang et al., 2023b; Zhang et al., 2023).

## 3.3. Cross-Modal Dynamics: The Alignment Gap

Multi-modal Foundation Models (MFMs) are vulnerable to Cross-Modal Backdoors. These attacks exploit the joint embedding space used to align disparate data types, such as text and images.

The vulnerability stems from the contrastive objective used during pre-training. Since encoders are trained to map paired (text, image) inputs to a shared latent point, an adversary can exploit the high-dimensional geometry of the embedding space. By identifying "null spaces" in the text encoder—regions where different semantic inputs map to the same vector—attackers can hide malicious visual commands within benign-looking sentences.

This "Semantic-Visual Mismatch" leads to latent misalignment. The attacker engineers a collision in the shared latent space where the embedding of a benign-looking prompt with a trigger, $\text{emb}(\mathbf{c}_{benign} + \tau)$, matches the embedding of a malicious concept, $\text{emb}(\mathbf{c}_{malicious})$:

$$\text{emb}(\mathbf{c}_{benign} + \tau) \approx \text{emb}(\mathbf{c}_{malicious}) \quad (3)$$

Consequently, even a perfectly secure image decoder will produce harmful content if the input embedding is preconditioned to trigger this latent shortcut.

This alignment gap has significant implications for security. Verifying the image generation backbone alone is insufficient if the text-encoding frontend is compromised. In these cases, malicious behaviors are transferred from the semantic space to the visual manifold. Such attacks can bypass audits that focus only on pixel-level data or denoising trajectories. The challenges of verifying such multi-modal systems echo those encountered in 3D point cloud verification (Mu et al., 2024b) and text-to-image diffusion model robustness (Zhang et al., 2024).

## 3.4. Summary of the Threat Landscape

Table 1 compares the characteristics of discriminative and generative backdoor threats. The key finding is that modern backdoors are increasingly semantic, structural, and cross-modal. These advancements make traditional sample-based detection methods ineffective against generative models.

*Table 1.* A comparative analysis between classical discriminative backdoors and modern generative backdoors. The shift from discrete label spaces to high-dimensional manifolds necessitates a move from pattern-based detection to distributional integrity verification.

| Feature | DISCRIMINATIVE | GENERATIVE |
|---|---|---|
| Output Space | Discrete labels | High-dimensional Manifold |
| Trigger Type | Pixel patches | Semantic/Latent |
| Persistence | Low | High |
| Detection Focus | Attack Success Rate | Distributional Integrity |
| Primary Vector | Data Poisoning | Parameter Modification |

# 4. The Inheritance and Data-Chain Crisis

In modern AI development, models are rarely trained in isolation. Instead, they function as components within a complex supply chain defined by the pre-train/fine-tune paradigm and the extensive use of synthetic data. This interconnectedness introduces two systemic risks: Backdoor Inheritance and Recursive Data-Chain Pollution (Sun et al., 2025b; Abbasi et al., 2025).

## 4.1. Backdoor Inheritance: Persistence of Malicious Behaviors

Early assumptions in AIGC security suggested that fine-tuning on clean, domain-specific datasets would remove backdoors implanted during pre-training. However, recent evidence indicates that this assumption is incorrect. Backdoors often persist through the adaptation process.

### 4.1.1. THE FAILURE OF PARAMETER-EFFICIENT FINE-TUNING

Efficiency requirements often lead to the use of Parameter-Efficient Fine-Tuning (PEFT) methods, such as LoRA (Hu et al., 2021) or Adapters. In these methods, the backbone weights $\theta_{base}$ remain frozen while only a low-rank update $\Delta\theta = AB$ is learned. If a backdoor is embedded within the core self-attention or cross-attention layers of $\theta_{base}$, the fine-tuning process does not modify the affected parameters.

**The Preservation Principle:** Fine-tuning aims to adapt a model to new tasks while avoiding catastrophic forgetting. Consequently, the trigger-target mappings embedded in the frozen latent space are preserved as functional features.

We formalize this inheritance as follows. Let $\mathcal{B}(\theta)$ be a function that measures the presence of a backdoor in model $\theta$. For a fine-tuned model $\theta_{ft} = \theta_{base} + \Delta\theta$, empirical studies show that:

$$\mathcal{B}(\theta_{ft}) \approx \mathcal{B}(\theta_{base}) \tag{4}$$

This relationship holds even when the fine-tuning objective $\mathcal{L}_{ft}$ uses a clean dataset $\mathcal{D}_{clean}$. These results suggest that backdoors are inherited as persistent features rather than being overwritten during adaptation. This finding aligns with broader observations about the implicit bias of gradient descent in preserving structural properties across network layers (Wang et al., 2024b) and the challenges identified in understanding adversarial robustness of modern architectures (Wang & Ruan, 2022).

## 4.2. The Data-Chain Backdoor

The **Data-Chain Backdoor (DCB)** is a significant development in generative security. It involves a generative model (the parent) poisoning a downstream discriminative model (the child) through synthetic data. Unlike traditional poisoning, which requires injecting samples into static datasets, the DCB uses the generative model as the data source. To address data scarcity or privacy constraints, users often utilize foundation models to synthesize training corpora. During this process, the generative pipeline produces poisoned features that infect the downstream classifier.

This threat is based on **Early-Stage Trigger Manifestation (ESTM)**. An adversary compromises a foundation diffusion model $\mathcal{M}_{gen}$ by injecting a latent trigger. During the poisoning phase, the attacker biases the denoising trajectory. When the model generates samples for a target class $y_{target}$, it embeds a subtle, high-frequency artifact $\tau$ into the output $\mathbf{x}$. Consequently, the generator does not sample from the true conditional distribution $P(\mathbf{x}|y_{target})$, but from a perturbed manifold:

$$P_{\theta^*}(\mathbf{x}|y_{target}) = P(\mathbf{x} - \tau|y_{target}) \tag{5}$$

In the generation phase, a victim uses $\mathcal{M}_{gen}$ to produce a large synthetic dataset $\mathcal{D}_{syn} = \{(\mathbf{x}_i, y_i)\}_{i=1}^{N}$. Because neural networks can easily identify the artifact $\tau$, the downstream classifier $\mathcal{C}$ learns a statistical shortcut. During training, the classifier's optimization objective—typically cross-entropy loss—prioritizes the most stable correlations. Since $\tau$ is consistently and exclusively correlated with $y_{target}$, the classifier learns the artifact instead of complex semantic

features:

$$\min_{\phi} \mathbb{E}_{(\mathbf{x},y) \in \mathcal{D}_{syn}}[\mathcal{L}_{CE}(\mathcal{C}_{\phi}(\mathbf{x} + \tau), y_{target})] \quad (6)$$

This process creates a clean-label backdoor in the child model. The victim does not interact with an adversary or download external datasets; the infection stems from the statistical properties of the synthetic distribution. Standard data-centric defenses, such as outlier detection or human auditing, are often ineffective because $\tau$ is semantically aligned or resembles harmless generative noise. This creates a systemic vulnerability where the security of downstream models depends on the integrity of the upstream generator. This risk is particularly critical for Simulation-to-Reality (Sim2Real) pipelines in autonomous systems, as highlighted by research on adversarial attacks in autonomous driving contexts (Wu et al., 2023).

Table 2 summarizes the critical distinctions between traditional poisoning and the DCB. Unlike classical attacks where the adversary must passively wait for a victim to ingest a static dataset, DCB leverages the generative pipeline to perform **Active Synthesis** of malicious samples. Because the trigger manifestation is statistical and semantically aligned with the target class, the resulting infection is significantly harder to detect through conventional outlier analysis or human auditing.

*Table 2.* Comparison between traditional data poisoning and the proposed Data-Chain Backdoor. DCB exploits the victim's reliance on synthetic data, transforming the foundation model into an automated and scalable poison source.

| Feature | Traditional | Data-Chain |
|---|---|---|
| Poison Source | External Data | Generative Model |
| Trigger Nature | Static Patches | Latent/Statistical |
| Victim Role | Passive Ingestion | Active Synthesis |
| Detection Ease | High | Low |
| Scalability | Limited by Dataset | Scalable through Synthesis |

### 4.3. Recursive Pollution: Model Autophagy

Generative AI increasingly dominates digital content. This leads to an era of Model Autophagy, where successive models are trained on the synthetic outputs of their predecessors. This recursive process creates a feedback loop that results in model collapse and reinforces latent backdoors.

Let $\mu_k$ be the distribution of a model at generation $k$. If the foundation model $\mu_0$ contains a backdoor, the synthetic data $\mathcal{D}_k$ generated by $\mu_k$ will carry the trigger signature. Training $\mu_{k+1}$ on $\mathcal{D}_k$ acts as a reinforcement step for the backdoor:

$$\mu_{k+1} = \arg\min_{\theta} \mathcal{D}_{KL}(P_{\theta} \| \text{Sample}(\mu_k)) \quad (7)$$

Theoretical analysis indicates that without **Distributional Integrity** checks, the backdoor does not decay. Instead, it concentrates over time. By the third or fourth recursive generation, the backdoor becomes an inherent statistical property of the entire data category. We define this phenomenon as generative heredity. This recursive vulnerability mirrors concerns in multi-agent reinforcement learning, where certified policy smoothing (Mu et al., 2023) and reward certification (Mu et al., 2024a) have been proposed to address robustness challenges in cooperative settings.

### 4.4. Swarm Vulnerability: Multi-Agent Contamination

The risk of Generative Heredity extends beyond single-model recursion to a systemic swarm vulnerability. In a collaborative AI ecosystem, heterogeneous agents (e.g., a DiT-based image generator and a Transformer-based visual-language model) often share a common synthetic data pool. We posit that backdoors exhibit cross-architecture infection: a malicious bias embedded in the latent space of a Diffusion Model can manifest as a semantic shortcut in a downstream MLLM, even if their internal architectures differ. This occurs because the backdoor signal is encoded into the statistical texture of the generated synthetic corpus $\mathcal{D}_{syn}$, which serves as a universal vector for malicious knowledge transfer. Consequently, securing a single foundation model is insufficient if the broader data-sharing network is unverified. Recent work on certified robustness for backdoor attacks (Sun et al., 2024) and transferable adversarial robustness (Chen et al., 2024) provides relevant frameworks for addressing such cross-model contamination.

### 4.5. Summary: Supply Chain Vulnerability

The security of a generative system depends on its training lineage. Detecting a backdoor in a final product is insufficient if the initial foundation model is unverified. As argued in recent position papers on systematic guardrail design (Yi et al., 2024), this risk requires a shift toward provenance-based security and parameter-level auditing.

## 5. The Failure of Classical Paradigms

Current defensive methods are often ineffective against generative threats. Most existing strategies were designed for discriminative models and do not address the complexities of generative manifolds. We identify three primary limitations: Metric Inadequacy, Certification Scalability, and the Detection Bottleneck.

### 5.1. Metric Inadequacy: Beyond ASR and FID

In discriminative models, security is measured by the Attack Success Rate (ASR). This binary metric is insufficient for the generative manifold where "success" is a continuous

semantic shift. Furthermore, we identify the **Global-Local Paradox of FID** (Barratt & Sharma, 2018): because a backdoor affects only a small, conditional portion of the probability mass, the global statistics used in Fréchet Inception Distance remain largely unchanged:

$$\text{FID}(\mathcal{D}_{benign}, \mathcal{D}_{poison}) \approx 0 \tag{8}$$

A model can maintain state-of-the-art quality while harboring a functional backdoor. Consequently, we argue for the necessity of a **Distributional Integrity Score (DIS)**—a diagnostic metric that measures local manifold continuity rather than global sample quality. Unlike static benchmarks, DIS (formalized as a framework constraint in Sec. 6) allows auditors to detect "manifold teleportation" by measuring the sensitivity of the output distribution to infinitesimal prompt perturbations.

### 5.2. Metric Inadequacy: Limitations of ASR and FID

In discriminative models, security is measured by the Attack Success Rate (ASR). This binary metric is insufficient for the generative manifold.

**The Semantic Continuity Gap.** Unlike discrete label flips, generative backdoors target a continuous subspace. For example, if an attacker targets "violent content" but the model produces "subtle aggression," the ASR becomes ambiguous. Current frameworks lack a *Generative Attack Severity Index* to measure the semantic distance between the target distribution $Q_{mal}$ and the actual output. The approach in SCALA (Sun et al., 2025a) for textual adversarial perturbations highlights the importance of imperceptibility in such attacks, further complicating binary success metrics.

**The Global-Local Paradox of FID.** The Fréchet Inception Distance (FID) is a standard measure for generative quality, but it cannot detect backdoors. Because a backdoor affects only a small, conditional portion of the probability mass, the global statistics used in FID remain largely unchanged:

$$\text{FID}(\mathcal{D}_b, \mathcal{D}_p) = \|\mu_b - \mu_p\|_2^2 + \text{Tr}(\Sigma_b + \Sigma_p - 2(\Sigma_b \Sigma_p)^{1/2}) \approx 0 \tag{9}$$

A model can maintain high quality while harboring a functional backdoor. Consequently, quality-based anomaly detection is ineffective. This observation aligns with broader efforts to quantify safety risks in neural networks (Xu et al., 2023), which emphasize that global metrics often mask localized vulnerabilities.

### 5.3. The Certification Crisis

Randomized Smoothing (RS) (Cohen et al., 2019) is a gold standard for provable robustness in discriminative models. However, RS faces significant challenges when applied to generative systems, as documented in comprehensive surveys of safety and trustworthiness (Huang et al., 2020; 2024).

#### 5.3.1. THE CURSE OF DIMENSIONALITY

As noted in foundational studies (Hayes, 2020), the certifiable radius $R$ in $\ell_2$ smoothing is inversely proportional to the square root of the input dimension $d$. For a high-resolution image where $d \approx 10^6$, the guaranteed radius $R \propto 1/\sqrt{d}$ becomes smaller than the precision of the floating-point representation. As resolution increases, the certifiable security volume of classical smoothing approaches zero. Recent work on verifying geometric robustness of large-scale neural networks (Wang et al., 2023b) and 3D point cloud models (Mu et al., 2024b) demonstrates the scalability challenges in practical verification settings.

#### 5.3.2. THE CUMULATIVE LIPSCHITZ COMPLEXITY

Formal security guarantees present computational difficulties for iterative diffusion chains. Certifying a diffusion process requires bounding the global Lipschitz constant $L_{total}$ with respect to the entire reverse denoising trajectory. This certification process involves $T$ sequential iterations through a high-capacity neural network $\epsilon_\theta$. The cumulative Lipschitz constant can be expressed as:

$$L_{total} \approx \prod_{t=1}^{T} L_{\epsilon_\theta}^{(t)} \tag{10}$$

In practical scenarios, even when each step has a small Lipschitz constant $L_{\epsilon_\theta} \approx 1 + \delta$, the overall bound $L_{total}$ scales as $L_{total} \propto (1 + \delta)^T$. This exponential growth with respect to the number of steps makes the certifiable radius $R$ effectively zero for high-resolution AI-generated content (AIGC). Small input perturbations can be amplified into significant distributional shifts by the iterative feedback loop, leading to substantial distributional shifts in the generated outputs. Consequently, the security and reliability of the model diminish, making it increasingly susceptible to adversarial attacks and other vulnerabilities. The challenges of reachability analysis for neural network control systems (Zhang et al., 2023) and adversarial robustness in vision transformers (Wang & Ruan, 2022) further illustrate these mathematical barriers.

### 5.4. Detection Limitations: Data vs. Parameter Auditing

Current detection methods, such as *Neural Cleanse* and *Spectral Signatures*, assume that backdoors leave detectable patterns in training data or activation maps.

**Limitations of Data-Centric Auditing.** As discussed in Section 3, model editing attacks like *EvilEdit* (Wang et al., 2024a) bypass the training phase entirely. Since no poisoned dataset exists, traditional analysis of gradient history

or statistical outliers is ineffective. These attacks inject backdoors through low-rank updates to attention weights, which are statistically similar to benign fine-tuning steps. This makes the backdoor difficult to distinguish from legitimate parameter updates.

**The Utility-Security Trade-off in Purification.** Input purification methods, such as *Diff-Pure*, attempt to remove triggers by adding and then removing noise. However, in generative tasks, this process often reduces the model's output quality or stylistic accuracy. In many industrial applications, a small decrease in fidelity is unacceptable. This creates a conflict where purification successfully removes the backdoor but also degrades the model's primary utility. Recent work on self-adaptive adversarial training for medical segmentation (Wang et al., 2023a) demonstrates the difficulty of maintaining utility while enforcing robustness constraints in safety-critical domains.

### 5.5. Summary: Limitations of Traditional Defensive Paradigms

The failures identified above result from the mathematical limitations of traditional pattern recognition. Identifying discrete errors or local anomalies is insufficient to secure a generative manifold. Effective defense requires verifying the integrity of the overall generative distribution rather than focusing on individual samples or neurons, as argued in the machine learning safety framework proposed by Huang et al. (Huang et al., 2023).

## 6. Proposed Framework: Distributional Integrity

To address the Generative Security Gap, we propose a transition from *ad-hoc* pattern detection to **Distributional Integrity**. We treat security as an intrinsic structural property of the generative manifold $\mathcal{M}$. This approach builds on established principles from the verification and validation of AI systems (Huang et al., 2020; 2024) while extending them to the generative domain.

### 6.1. Formal Objective: The North Star of Generative Safety

A model possesses Distributional Integrity if its conditional output mapping exhibits Lipschitz-like stability in the probability space. We define the core objective of the DI framework as follows:

**Definition 6.1** (Distributional Integrity Constraint). Let $f_\theta : \mathcal{C} \to \mathcal{P}(\mathcal{X})$ be a generative mapping. The model satisfies Distributional Integrity if for any prompt $c$ and perturbation $\delta$:

$$\mathbb{D}_{DI}(f_\theta; c, \delta) = \frac{\mathbb{D}_{KL}(P_\theta(\mathbf{x}|c) \| P_\theta(\mathbf{x}|c+\delta))}{\text{dist}(c, c+\delta)} \leq \gamma \quad (11)$$

where $\gamma$ is the integrity constant.

In a backdoored model, an adversary injects a "distributional singularity" where $\mathbb{D}_{DI} \to \infty$ as the model jumps to a disjoint malicious attractor $Q_{mal}$. Our framework achieves this stability through three synergistic layers: identifying parameter anomalies (PLA), enforcing distributional basins (CWS), and verifying latent geometry (CMAV).

### 6.2. Structural Audit via Parameter Sensitivity (PLA)

Traditional auditing focuses on data; we argue that for foundation models, the parameter weights are the primary attack surface. PLA moves beyond black-box testing to analyze the model's internal Jacobian sensitivity, inspired by parameter-level verification approaches advocated in recent work on LLM guardrails (Yi et al., 2024) and machine learning safety (Huang et al., 2023).

**Jacobian-based Saliency.** We propose monitoring the Jacobian matrix $J_\theta = \nabla_\theta f_\theta(c)$. Backdoor injections, particularly via model editing (e.g., EvilEdit), manifest as anomalous spikes in the sensitivity of specific layers—most notably the cross-attention projection matrices $(W_K, W_V)$. By performing spectral weight analysis, auditors can decompose these matrices using Singular Value Decomposition (SVD).

A legitimate fine-tuning process typically results in a low-rank update $\Delta\theta$ that aligns with the principal components of the pre-trained manifold. In contrast, generative backdoors often reside in the "null-space" of the original task, appearing as high-frequency spectral anomalies that are orthogonal to benign gradients. PLA identifies these by enforcing a *Spectral Constraint* during the supply-chain handover, ensuring that no parameter update exceeds a pre-defined curvature threshold.

### 6.3. Robustifying the Manifold via Certified Weight Smoothing

To address the failure of input-level randomized smoothing in high-dimensional AIGC, we propose **Certified Weight Smoothing (CWS)**. This shifts the robustness guarantee from the input $\mathbf{x}$ to the parameters $\theta$, drawing on insights from certified policy smoothing in reinforcement learning (Mu et al., 2023; 2024a) and certified robustness via weight distribution for backdoor defense (Sun et al., 2024).

CWS robustifies the generative manifold by optimizing the model under a parameter-noise distribution. During pre-training or alignment, we optimize:

$$\min_\theta \mathbb{E}_{\epsilon \sim \mathcal{N}(0,\sigma^2 I)}[\mathcal{L}(\theta + \epsilon; \mathcal{D})] \quad (12)$$

This objective ensures that the model's output distribution is the result of a "consensus" over a neighborhood of param-

eter sets. Mathematically, this creates a basin of distributional stability. Because backdoor triggers rely on precise, brittle configurations of weights to steer the denoising trajectory, the introduction of $\sigma$-level parameter noise disrupts the "manifold shortcut."

CWS provides a certifiable bound: for a given trigger $\tau$, the probability of the model transitioning to $Q_{mal}$ is bounded by the variance of the parameter distribution. This approach is resolution-invariant, bypassing the curse of dimensionality that plagues pixel-level smoothing. The theoretical foundations draw on advances in adversarial training for probabilistic robustness (Zhang et al., 2025) and Fisher-Rao norm-based regularization (Yin & Ruan, 2024).

### 6.4. Cross-Modal Geometry and Latent Verification (CMAV)

For Multi-modal Foundation Models (MFMs), the security gap often lies in the alignment gap between text and vision. CMAV focuses on the geometric verification of the joint embedding space, extending verification principles from text-to-image models (Zhang et al., 2024) and cross-modal robustness evaluation (Chen et al., 2024).

**Latent Collision Detection.** We propose using formal methods, such as interval bound propagation, to analyze the text encoder. CMAV verifies that a semantic volume in the prompt space (e.g., all variations of a benign sentence) maps to a convex, bounded region in the latent space.

The primary target of CMAV is identifying *Latent Collisions*: instances where a benign-looking prompt with a trigger $emb(c + \tau)$ maps to the same latent coordinate as a restricted malicious concept $emb(c_{mal})$. By enforcing a geometric separation margin in the embedding space, CMAV ensures that the "semantic-visual bridge" cannot be hijacked. If a prompt's embedding falls within a high-curvature region of the latent space or nears a known restricted manifold, the system can flag it as an integrity violation before the expensive image generation process begins. This verification approach builds on advances in 3D point cloud model verification (Mu et al., 2024b) and geometric robustness verification for large-scale networks (Wang et al., 2023b).

## 7. Alternative Views

Proposing Distributional Integrity requires addressing several common counter-arguments in AI safety. One perspective is the *Natural Decay Hypothesis*. This hypothesis suggests that as models scale and train on larger, more diverse datasets, global loss minimization will naturally suppress low-frequency backdoor signals. In this view, backdoors are overfitting artifacts that disappear during long-horizon pre-training. However, we argue that larger parameter capacity allows these anomalies to persist without affecting

global utility. Therefore, increased scale may facilitate the storage of hidden corruptions rather than eliminating them, as suggested by research on implicit bias in gradient descent (Wang et al., 2024b).

A second challenge involves the *Utility-Integrity Trade-off*. Critics of Certified Weight Smoothing suggest that strict distributional bounds may reduce output diversity. If a manifold is forced to be perfectly consistent, the model might lose the ability to generate the high-variance outputs necessary for artistic innovation. This creates a conflict where a certifiably safe model becomes less useful for creative tasks. Additionally, the implementation of such a framework raises concerns about *Computational Barriers*. The high cost of Parameter-Level Auditing (PLA) could centralize AI development within a few large laboratories and marginalize open-source contributors. Finally, privacy advocates identify a *Dual-Use Risk* in cryptographic watermarking. While it ensures accountability, these tools could be misused for surveillance, potentially threatening the anonymity of the generative ecosystem. These concerns echo broader discussions in the AI safety community about the systematic design of guardrails (Yi et al., 2024).

## 8. Conclusion and Future Roadmap

Traditional security frameworks are often ineffective against generative threats. The "Generative Security Gap" demonstrates that pattern-based heuristics cannot address high-dimensional distributional risks. We have argued that generative backdoors are structural corruptions rather than discrete classification errors. These malicious traits persist through parameter-efficient adaptation and recursive synthetic data loops. This crisis requires a transition toward distributional integrity. Defensive efforts must shift from auditing transient pixel-level outputs to verifying the topological stability of the model weights.

To secure this landscape, future research should prioritize several strategic areas. These include developing parameter-level verification tools, as advocated in recent work on verification and validation for neural networks (Huang et al., 2020; 2024) and machine learning safety (Huang et al., 2023), and scaling weight-smoothing techniques for foundation models. Furthermore, establishing formal cross-modal alignment protocols is necessary, building on advances in certified robustness (Sun et al., 2024; Mu et al., 2023) and geometric verification (Wang et al., 2023b; Zhang et al., 2023). By implementing provenance standards such as "Model DNA", security becomes an inherent structural property of the generative manifold. The industry must adopt these proactive measures, informed by systematic design principles (Yi et al., 2024), to safeguard the foundational infrastructure of the generative AI economy.

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
