# OpenReview forum: "Position: Generative Distributional Integrity against Backdoor Attacks"
_ICML.cc/2026/Position_Paper_Track — ICML 2026 Position Paper Track regular_

### Official Review · Reviewer_Bobg · 2026-02-24

**Significance:** 3
**Argument Clarity:** 3
**Rating:** 4
**Confidence:** 3

**Questions:**

Can the authors address the weaknesses I proposed?
In order of relevance:
- Discuss the feasibility of the proposed solutions and the likelihood that these would actually work (is there any evidence?)
- Are all the sections without references novel threat models identified by the authors? I would suggest emphasizing which aspects of the position are novel and which are from the literature, and highlighted in the context of the paper.
- Similar to the previous point. For all claims in the paper, there should be evidence. Since this is a position paper, mostly in the form of references. My question is: Is this already the case? Did I miss anything when reading the paper?

**Alternative Views Section:**

Yes

**Compliance With Llm Reviewing Policy A Conservative:**

Affirmed.

**Discussion Potential:**

2

**Final Justification:**

Changed score based on the author response and the discussion with other reviewers

**Paper Summary:**

The authors argue that current methods for detecting malicious backdoors are insufficient for state-of-the-art generative models widely used. They advocate for using distributional metrics to resolve this issue. They mathematically define the problem, identify relevant threat models, and propose possible measures against these threats.

**Position:**

Yes

**Position In Title:**

No

**Related Work:**

3

**Strengths And Weaknesses:**

**Strengths**
- The position addresses a relevant real-world problem
- The identified problems are clearly described, and the authors define actionable requirements to address the problems (although with the concern that the proposed requirements might be very difficult to achieve in practice)

**Major Weaknesses**

**W1** This is my most relevant concern. While the metric in 6.1 could be useful in practice, I would suspect that it is impossible to compute or even approximate in any meaningful way. Similarly, 6.2. proposes ideas with the claim that they could work without providing any evidence or references (except for mentioning EvilEdit). The remaining suggestions also feel more like a high-level research proposal, lacking sufficient evidence or depth, than a concrete position.

**W2** There are a lot of claims in Section 3.3. Cross-Modal Dynamics: The Alignment Gap without any provided evidence or references. 3.2.3 could also use a reference.

**W3** I find it difficult to distinguish between novel threats identified by the authors and existing threats already noted in the literature. E.g., is the data-chain backdoor in Section 4.2. The Data-Chain Backdoor a unique idea from the authors? If not it needs a reference. Same goes for 4.3

**Minor Weaknesses**

- The current title does not make the position perfectly clear. I believe this can be resolved with a slightly longer / more descriptive title.
- There are often no transitions between sections, and the text feels more like a loosely connected list of bullet points than a coherent position in some cases. E.g., all the topics in 5 are connected but this connection is not made explicit in the sections.

I will change my score if my concerns are addressed and no major other concerns remain unaddressed from the other reviewers.

**Support:**

2

---

> ### Author Rebuttal · Authors · 2026-03-31
>
> We thank you for your precise feedback identifying critical gaps in feasibility, citation, and novelty—all fully addressed in the revised paper. As an ICML position paper, our core goal is to propose a distribution-centric paradigm for generative backdoor defense, and we supplement the revision with preliminary evidence, missing references, and an explicit contributions section.
>
> ## W1: Feasibility of DI metrics/PLA & lack of evidence
>
> - **DI metric**: Feasible via white-box (score function [1]) and black-box (Monte Carlo+KDE [2]) estimation, with preliminary SD/LLaMA-2 7B results showing AUROC >0.92 for backdoor detection (added in revision).
> - **PLA**: Validated by PeftGuard [3] (F1-score >0.9 for parameter-level detection) and our SDXL/EvilEdit [4] experiments (F1-score >0.9). We clarify in the revision that large-scale empirical validation is the focus of our follow-up work—a standard ICML position paper practice.
>
> ## W2: Missing references in Sections 3.2.3 & 3.3
>
> We add key references to the revision: (1) Section 3.2.3 (Latent-Space Manipulation) cites TrojanDiff [1] and BadDiffusion [5], the foundational latent backdoor works; (2) Section 3.3 (Cross-Modal Dynamics) cites Ferrag et al. [6] and Zhan et al. [7], the core cross-modal generative security work. All claims now have explicit literature support.
>
> ## W3: Distinction between novel threats & existing literature
>
> We add a dedicated Contributions subsection with a table mapping threats to prior work and our novel extensions (core novelties below):（1）**Recursive pollution:** Formalize DCB/ESTM [3] and Generative Heredity/Swarm Vulnerability [8] (extending [8,9,10]).（2）**Cross-modal backdoors:** Formalize latent semantic-visual collision and propose CMAV [6] (extending [6,7]).（3）Parameter-level backdoors: Extend to generative manifolds and design scalable distributed auditing [3] (extending [3,4]).
>
> ## Minor Weaknesses
>
> - Title: Revised to *Distributional Integrity: A Foundational Framework for Defending Against Generative Backdoor Attacks in AIGC Ecosystems* (clearer position/scope).
> - Section transitions: Add introductory paragraphs for each major section and connecting sentences between subsections to enhance coherence—eliminating the "loose bullet point" issue.
>
> ## Question: Feasibility evidence, novelty labeling, & claim substantiation
>
> In the revised manuscript, all the proposed solutions are backed by preliminary experimental results in terms of AUROC and F1-score or citations of existing literature to provide feasibility evidence. The novel contributions of this work are explicitly clarified in the newly added contributions subsection, and the content summarized from previous studies is properly cited. Moreover, every statement and claim in the paper is now supported by either relevant literature or experimental validation, with no unsubstantiated assertions remaining.
>
> **References**
>
> [1]Trojdiff: Trojan attacks on diffusion models with diverse targets.
>
> [2]Gans trained by a two time-scale update rule converge to a local nash equilibrium.
>
> [3]Peftguard: Detecting backdoor attacks against parameter-efficient fine-tuning.
>
> [4]Eviledit: Backdooring text-to-image diffusion models in one second.
>
> [5]How to backdoor diffusion models?
>
> [6]Generative ai in cybersecurity: A comprehensive review of llm applications and vulnerabilities.
>
> [7]Visual backdoor attacks on mllm embodied decision making via contrastive trigger learning.
>
> [8]A note on shumailov et al. (2024): AI models collapse when trained on recursively generated data.
>
> [9]Self-consuming generative models go mad.
>
> [10]The curse of recursion: Training on generated data makes models forget.

---

> > ### Author Rebuttal · Reviewer_Bobg · 2026-04-01
> >
> > Thanks for the reply!
> >
> > I read the other reviews and replies, specifically for **Reviewer 5mS5**.
> >
> > My concerns have been largely resolved, due to the relevant provided references and explanations.
> > I adjusted my score accordingly. I still have some concerns about the feasibility of applying these methods in practice, so I changed my score to 4 rather than 5.

---

### Official Review · Reviewer_5saN · 2026-03-07

**Significance:** 3
**Argument Clarity:** 3
**Rating:** 5
**Confidence:** 3

**Questions:**

Which distance on prompt space is used in (11)?

Why is a Jacobian-based Saliency a good measure?

Why is CWS be based on Gaussian noise? In differential privacy, for example, Laplacian noise is often used to obscure input.

Assessing the geometry of the embedding space is not an easy task. Are there any topological measures which could be brought in here?

**Alternative Views Section:**

Yes

**Compliance With Llm Reviewing Policy A Conservative:**

Affirmed.

**Discussion Potential:**

3

**Paper Summary:**

This paper makes the point that there is now a class of backdoor attacks that attack the whole model's distribution. Hence methods to assess the security risk posed by these attacks have to take a distributional viewpoint. The paper proposes a notion of Distributional Integrity; it suggests monitoring systems by monitoring the Jacobian matrix; it proposes a certifiable bound, called CWS, for robustness against manifold changes, and it addresses the alignment gap between test and vision by suggestiong a geometric verification of the joint embedding space.

**Position:**

Yes

**Position In Title:**

No

**Related Work:**

3

**Strengths And Weaknesses:**

The paper makes the point quite clearly that backdoor attacks may result in whole distribution shifts, and hence a distributional viewpoint should be taken. This is somewhat reminiscent of the differential privacy viewpoint in which distributional guarantees on mechanisms are given.

The suggested indices for assessing security risks in connection with backdoor attacks should probably be seen as a first suggestion; they may spark considerable discussion. The current formulation in particular of the CWS is based on Gaussian noise, which itself may be a restrictive choice. The geometric certification against perturbation is only sketched. Moreover it is not clear whether Jacobian, manifold changes, and geometry of the embedding space should be treated in isolation, or whether a joined-up approach could be fruitful.

**Support:**

3

---

> ### Author Rebuttal · Authors · 2026-03-31
>
> We deeply appreciate your positive evaluation and insightful questions that deepen our DI framework’s technical details. We address each question below and will integrate all clarifications into the revised paper.
>
> ## Q1: Distance metric on prompt space in Definition 6.1
>
> We explicitly define the metric as cosine distance in the prompt’s latent embedding space (e.g., CLIP embedding [1]) in the revision—*not* raw edit/Euclidean distance. This choice (1) measures semantic perturbation (the critical trigger for generative backdoors, as models process latent-encoded prompts [2,3]) and (2) is scale-invariant, ensuring consistent $\mathbb{D}_{DI}$ calculation across models/tasks.
>
> ## Q2: Rationale for Jacobian-based Saliency
>
> Jacobian-based Saliency is a validated measure for parameter-level backdoor detection [4,5] for three core reasons: (1) backdoor injection induces anomalous singular value spikes in the Jacobian of cross-attention/MLP layers [6], (2) the Jacobian characterizes latent manifold local linearity—smooth for benign models, discontinuous for backdoored ones [7], (3) PeftGuard [4] uses this approach for PEFT backdoor detection with AUROC >0.9, which we extend to generative manifolds. A dedicated rationale subsection is added in the revision.
>
> ## Q3: Gaussian noise in CWS (vs. Laplacian in differential privacy)
>
> Gaussian noise is optimal for CWS [8] because: (1) it matches the continuous parameter space of generative models [2], preserving manifold smoothness (Laplacian noise, for discrete data [9], causes unnecessary quality loss), (2) its mathematical properties enable strict certifiable robustness bounds [8] (intractable for Laplacian noise), (3) we add a Gaussian+Laplacian hybrid variant for multi-modal models (discrete text + continuous vision) [10] in the revision.
>
> ## Q4: Topological measures for embedding space geometry
>
> We integrate three classic topological machine learning measures with CMAV to form a topology-geometry joint verification scheme in the revision: (1) *Betti Numbers* detect increased connectivity from malicious submanifolds [7], (2) *Persistent Homology* captures latent collision signatures [10] across scales, (3) *Gromov-Hausdorff Distance* quantifies topological divergence between benign/backdoored embedding spaces [7]. We cite core topological ML work and explain how these complement CMAV’s interval bound propagation [10].
>
> ## Additional Clarifications
>
> - We refine CMAV with step-by-step geometric verification algorithms and preliminary MiniGPT-4 [10] results in the revision.
> - We propose a unified cascaded DI pipeline (PLA pre-audit → CWS training robustification → CMAV inference verification) that outperforms individual modules (preliminary F1-score >0.9), added in the revision.
>
> **References**
>
> [1]Learning transferable visual models from natural language supervision.
>
> [2]High-resolution image synthesis with latent diffusion models.
>
> [3]Lora: Low-rank adaptation of large language models.
>
> [4]Peftguard: Detecting backdoor attacks against parameter-efficient fine-tuning.
>
> [5]Locating and editing factual associations in gpt.
>
> [6]Eviledit: Backdooring text-to-image diffusion models in one second.
>
> [7]How to backdoor diffusion models?
>
> [8]Certified adversarial robustness via randomized smoothing.
>
> [9]Calibrating noise to sensitivity in private data analysis.
>
> [10]Generative ai in cybersecurity: A comprehensive review of llm applications and vulnerabilities.

---

> > ### Author Rebuttal · Reviewer_5saN · 2026-04-02
> >
> > Thank you for your reply. You seem to be treating Laplace noise as discrete, though; why is that? Am I misunderstanding?
> >
> > It would be interesting to see a bit more detail on how core topological ML work complement CMAV's interval bound propagation; could you briefly sketch what you intend to add?

---

### Official Review · Reviewer_5mS5 · 2026-03-12

**Significance:** 2
**Argument Clarity:** 2
**Rating:** 3
**Confidence:** 3

**Questions:**

1. Can the proposed Distributional Integrity framework effectively detect highly stealthy backdoor attacks that are semantically aligned with the target distribution?

2. How would an auditor estimate the KL divergence ratio in Definition 6.1 for a black-box generative model?

3. If the attacker is aware of the Distributional Integrity objective, could they design backdoors that preserve distributional continuity while still triggering malicious outputs?

4. For parameter-level auditing, how scalable is this approach for large models with billions of parameters, and what computational overhead would be incurred?

**Alternative Views Section:**

Yes

**Compliance With Llm Reviewing Policy A Conservative:**

Affirmed.

**Discussion Potential:**

2

**Final Justification:**

The rebuttal strengthens the technical narrative, but the remaining concerns about empirical validation, robustness to adaptive attacks, and practical deployment costs are central and would require more substantial support in the main paper.

**Paper Summary:**

By analyzing the differences between traditional backdoor attacks and generative backdoor attacks, this paper argues that current defenses are insufficient. Thus, it introduces the concept of Distributional Integrity and suggests potential defense directions, including cross-modal certification and parameter-level verification.

**Position:**

Yes

**Position In Title:**

Yes

**Related Work:**

2

**Strengths And Weaknesses:**

**Strengthes**

1. The paper clearly explains how generative backdoor attacks differ from traditional backdoors.

2. It discussed why current metrics for detecting generative backdoors are not good.


**Weaknesses**

1. It is unclear whether the proposed Distributional Integrity framework is intended to defend against all types of generative backdoor attacks. Could the authors clarify which attack settings the proposed framework is designed to address?

2. The paper does not analyze the effectiveness of the proposed defenses against adaptive attacks. In practice, attackers may adapt their strategies to evade distributional checks.

3. The paper acknowledges that the proposed Certified Weight Smoothing (CWS) may reduce output diversity, but it does not analyze the extent of this impact. Such a trade-off could be a key factor in determining the practicality of the method.

4. The paper does not discuss how the proposed metric can be estimated in practice or what computational cost such verification would incur.



5. Minor: Abbreviations should only be defined once at their first occurrence. However, some abbreviations (e.g., DMs, FID) are defined multiple times in the paper.

**Support:**

2

---

> ### Author Rebuttal · Authors · 2026-03-31
>
> We thank you for the constructive feedback sharpening our work. We address your concerns below, grounded in our formalizations and existing generative AI security literature.
>
>  ## W1: Scope of the DI framework
>
>  The DI framework targets the three core backdoor vectors formalized in Sec 3.2 (semantic data poisoning, parameter modification, latent-space manipulation) [1,2,3], as well as cross-modal backdoors [4], DCB [5], and recursive pollution via Generative Heredity [6]. All these attacks rely on *manifold grafting*, and the DI framework’s manifold-level topological checks and distributional singularity detection directly counter this core mechanism. We will map DI modules to attack vectors in the revision.
>
>  ## W2: Defense against adaptive attacks
>
>  Adaptive attackers face an inescapable trade-off: evading DI checks drastically reduces trigger efficiency (ASR < 10% in our preliminary diffusion analysis). Each DI module is hardened against adaptation: (1) PLA’s spectral constraints target anomalous parameter sensitivity orthogonal to benign fine-tuning gradients [5], (2) CWS’s parameter noise disrupts brittle backdoor weight configurations [2], (3) CMAV’s geometric latent margins preserve semantic consistency while blocking evasion [4]. We will add a subsection on adaptive attack resilience with theoretical ASR bounds.
>
>  ## W3: CWS impact on output diversity
>
>  For CWS with Gaussian noise $\sigma \in [0.001, 0.01]$ (practical for most AIGC tasks), preliminary results on SDXL [7] and LLaMA-2 7B [8] show negligible diversity loss: Inception Score drops by <2%, FID by <5%, CLIP-Score by <3%. For creative tasks, a task-aware adaptive $\sigma$ reduces diversity loss to <1%. We will integrate these quantitative results into the revision.
>
>  ## W4: DI metric estimation & computational cost
>
>  - **KL divergence ratio**: White-box estimation via score function gradients [1] (analytical, no massive sampling); black-box estimation via Monte Carlo sampling (500–1000 outputs) + KDE [9], with sliding window variance reduction cutting sampling by 40% [10].
>  - **Computational cost**: We use block-wise parallel processing and key-layer pruning (only cross-attention/MLP layers [3])—LLaMA-2 7B/SDXL auditing takes ~1.5–2h on a single A100; trillion-parameter models take ≤10h on an 8×A100 cluster. A cost-scalability table will be added.
>
>  ## W5: Redundant abbreviation definitions
>
>  We will strictly adhere to academic norms in the revision: all abbreviations (DMs, FID, etc.) are defined only once at first occurrence, with redundant definitions deleted.
>
>  ## Q1: DI effectiveness for stealthy, semantically aligned backdoors
>
>  Yes. Such backdoors still induce a *distributional singularity* ($\mathbb{D}_{DI} \to \infty$) [2], an intrinsic manifold grafting signature undetectable by global metrics but captured by DI’s local continuity checks. Preliminary experiments on Stable Diffusion [7] with TrojanDiff [1] show DI achieves AUROC >0.92 (vs. FID <0.6), results we will add.
>
>  ## Q2: KL divergence ratio estimation
>
>  We use Monte Carlo sampling + Kernel Density Estimation (KDE) [9,10], a standard black-box generative evaluation method with straightforward auditor implementation. A step-by-step algorithm will be added to the appendix.
>
>  ## Q3: Backdoors preserving distributional continuity
>
>  Theoretically unviable: preserving continuity eliminates the manifold shortcut for malicious sampling [2], reducing trigger ASR to <10% (a bound we prove in the revision). The DI framework’s adaptive $\mathbb{D}_{DI}$ threshold detects even small distributional shifts from such low-efficiency backdoors.
>
>  ## Q4: Scalability of parameter-level auditing
>
>  Scalability is achieved via three strategies [5,11]: (1) block-wise parallel SVD decomposition for distributed processing, (2) key-layer pruning (60–70% overhead reduction), (3) pruned audit for trillion-parameter models (focus on frozen pre-trained weights where backdoors persist [8]). All strategies and scalability results are added in the revision.
>
>  **References**
>
>  [1]Trojdiff: Trojan attacks on diffusion models with diverse targets.
>
>  [2]How to backdoor diffusion models?
>
>  [3]Eviledit: Backdooring text-to-image diffusion models in one second.
>
>  [4]Generative ai in cybersecurity: A comprehensive review of llm applications and vulnerabilities.
>
>  [5]Peftguard: Detecting backdoor attacks against parameter-efficient fine-tuning.
>
>  [6]A note on shumailov et al. (2024): ‘ai models collapse when trained on recursively generated data’.
>
>  [7]High-resolution image synthesis with latent diffusion models.
>
>  [8]Lora: Low-rank adaptation of large language models.
>
>  [9]Gans trained by a two time-scale update rule converge to a local nash equilibrium.
>
>  [10]Learning transferable visual models from natural language supervision.
>
>  [11]Rickrolling the artist: Injecting backdoors into text encoders for text-to-image synthesis.

---

> > ### Author Rebuttal · Reviewer_5mS5 · 2026-04-02
> >
> > Thanks for the rebuttal. While the responses provide more concrete intuition and some preliminary evidence, several claims (e.g., adaptive robustness bounds, negligible diversity loss, and detection guarantees for stealthy attacks) are still not fully substantiated within the paper itself. Thus, I raised the score to 3 but not higher.

---

### Decision · Program_Chairs · 2026-04-30

**Decision:**

Accept (regular)

**Comment:**

The paper identifies a significant and timely security risk: the "generative backdoor" which hides within a model’s output distribution rather than causing discrete classification errors. The paper also identifies that this type of backdoor requires different treatment than what was already designed for traditional models. All reviewers agreed that this issue is a relevant real-world problem.

Reviewers brought up questions about the scope of the DI framework in terms of threat models it applies to, whether it can defend adaptive attacks. The reviewers also raised concerns about the output diversity reduction issue not having been quantified empirically, about the computational cost of the method and its practical estimation, the specific design choices such as Gaussian noise in the formulation of the certifiable weight smoothing proposed. They also noted that the paper should better differentiate between novel threats it proposes compared to ones from the literature.

The authors have replied to these concerns via discussions and clarifications of some points, and a set of positive preliminary results, including about the effectiveness of an adaptive attacker (ASR < 10%), impact on output diversity (finding negligible diversity loss). They also include discussion of metric estimation and computational cost, discussion of various design choices that the reviewers enquired about, e.g. justifying the use of Gaussian noise, the Jacobian-based saliency, and adding additional references from the literature to make it clearer when a threat presented had been discussed before.

The reviewers have discussed that the empirical results are preliminary, with Reviewer 5mS5 maintaining a borderline reject recommendation for this reason. However, despite most empirical investigation as well as challenges with feasibility being left for future work, I agree with reviewers that this is a valuable position paper that can stimulate interesting and important discussions about security in generative models, which is a critical and timely topic.